

# Physical laws for precursory phenomena of impending large earthquakes and their applications to predictions

Fumihide Takeda [1,2]

[1] Earthquake Prediction Institute at Imabari, 3–4–56 Go–Shin–Yashiki–Cho, Imabari, Japan
[2] Takeda Engineering Consultant Co., 2–14–23 Ujina Miyuki Minamiku, Hiroshima, Japan

*Correspondence to*:  Fumihide Takeda (takeda–f@epi21.org)

**Abstract.** Changes in crustal stresses create an earthquake fault motion which radiates seismic waves. Their analyses
quantify the properties of the earthquake with its rupture time, location, fault motion and size that are called earthquake
source parameters. We may then regard the event as the emergence of a virtual particle of unit mass at a position in the
property space whose coordinate axes are source parameters. At the next event, the particle takes a new position in the space.
The consecutive events draw a pathway of the moving particle, which is a trace of the stress changes causing the particle
motion. The pathway is zigzagged and non–derivative with respect to time. A mathematical tool named physical wavelets is
applied to extracting the equations of particle motion. The extracted equations detect periodic anomalous accelerations
precursory to large impending earthquakes weeks and months ahead of time. The periodic particle motions enable us to
predict the fault size and motion, rupture time, and epicenter of impending large earthquakes. The mathematical tool with
which to extract deterministic precursors embedded in the highly irregular time series of natural and earth systems is also
concisely described for mitigating their hazards.

## 1 Introduction

The earth lithosphere may be layered into three parts: the brittle (B) upper crust, the ductile (D) lower crust and the D upper
most mantle. The crustal stress state whose principal stress is expected normal to the Earth's surface with the other two
stresses acting in an approximately horizontal plane is as follows (Zoback and Zoback, 2002). The plate–driving force of
about $3\times10^{12}$ Nm$^{-1}$ creates steady state creep in the D parts. If the creep deformation rate is high, the stress in the B part
builds up and the region becomes tectonically active by coupling of the three layers. Earthquakes (EQ's) of various sizes
then occur in the B part.

The stress state may be quantified with the EQ events and the geological displacements provided with Global Positioning
System (GPS). The EQ events follow the changes in stress state; however, the occurrences are irregular by reflecting the
statistically self–similar (fractal) scaling of the events (Aki, 1981; Turcotte and Malamud, 2002). Faint determinism (leading
to large events) in the EQ events may be masked with their irregularities. The displacements of GPS stations are the changes
on crustal surface; however, every displacement time series shows large trends and environmental random noises that mask





subtle deterministic changes coupled with the plate motions like those precursory to the 2011 Tohoku M9 EQ (Takeda, 2011a, 2013, 2015; TEC21, 2017a).

In this article, we extract equations of precursory EQ motion from the EQ events with a mathematical tool named physical wavelets. As an example of many test and real–time extractions (Takeda and Takeo, 2004; Takeda, 2015; TEC21, 2017b),

we detailed the equations of the 1995 Kobe M7.2 event extracted two months before it occurred. The EQ equations are physical laws describing precursory phenomena of impending large earthquakes. They may be applicable to the short–term deterministic prediction of large and great EQ's (Takeda, 2015; TEC21, 2017b). The physics of the deterministic short–term predictions is explored with physical wavelets. The formulation and usage of physical wavelets are different from those in well-known wavelets analyses as described in section 3.

**2 Earthquake events as a virtual particle motion in a space**

A nation of EQ's has a dense seismic network laid out. If the network detects an EQ event, it assigns the EQ source parameters to the event (JMA, 2017; NIED, 2017). The parameters are the epicenter location (in latitude *LAT*, longitude *LON* and focal depth *DEP*), its origin time (event time), and magnitude *MAG*. The time interval between consecutive events is the inter–event interval (*INT*), which reflects changes in the stress state of the region (Dieterich, 1994).

We construct an EQ source parameter space with the *LAT*, *LON*, *DEP*, *INT* and *MAG* coordinate axes. An event, which may be regarded as the emergence of a virtual particle of unit mass, has a position in the EQ space. On the next EQ event, the particle will move to a new position. The continuous events draw the pathway of motion in space. Each *c*–coordinate (*c* = *LAT*, *LON*, *DEP*, *INT* and *MAG*) is the time series (Takeda and Takeo, 2004, 2007; Takeda, 2011b, 2015; TEC21, 2017c);

$$\{ c \} = \{ d(c,1), d(c,2), \cdots, d(c,m), \cdots \}. \qquad (1)$$

The $d(c, m)$ is the *c*–coordinate of the particle position at time *m*. As for the arrival time of the particle, it was the origin time of the EQ event; however, the origin time is uniquely related to the chronological event index *m*. So, the arrival time is replaced with the index *m* in $\{c\}$. The *INT* is not calculated at *m* = 0 so that the time of the particle motion in space starts from *m* = 1. The motion discontinuously changes its direction and speed so that the derivative of $d(c, m)$ with respect to time

*m* cannot be defined. We use physical wavelets to define the derivatives of such a highly irregular motion.

**3 Physical wavelets**

**3.1 Correct difference expression for derivatives**

Consider the motion of a virtual particle in the EQ space. We first assume the position of the particle is differentiable with respect to time *t* which is a real number. Denote the position by $D(c, t)$ and an interval of *t* by $\Delta t$ ($\geq 0$). We customarily define

the *c* component of the velocity,





$$V(c,t) = \lim_{\Delta t \to 0} [D(c, t + \Delta t) - D(c,t)] / \Delta t = \mathrm{d}D(c,t) / \mathrm{d}t. \qquad (2)$$

The differential operator d/d$t$ has the time reversal property of d/d($-t$) $= -$ d/d$t$. This time reversal of $-t$, while keeping the interval $\Delta t$ positive, changes the forward difference in Eq. (2) into, $[D(c, -(t-\Delta t)) - D(c, -t)] = -[D(c, t) - D(c, t-\Delta t)]$, for

which $D(c, -t) = D(c, t)$. The difference does not obey the time reversal of d/d$t$. The correct representation is then the central difference,

$$V(c,t) = \lim_{\Delta t \to 0} [D(c, t + \Delta t / 2) - D(c, t - \Delta t / 2)] / \Delta t = \mathrm{d}D(c,t) / \mathrm{d}t. \qquad (3)$$

Using the Dirac delta function $\delta(\tau)$, Eq. (3) is,

$$V(c,t) = \lim_{\Delta t \to 0} \{ \int_{-\infty}^{+\infty} D(c, \tau)[\delta(\tau - t - \Delta t / 2) - \delta(\tau - t + \Delta t / 2)] \mathrm{d}\tau \} / \Delta t. \qquad (4)$$

The $\delta(\tau)$ is an even function of time $\tau$ with the property of,

$$D(c,t) = \int_{-\infty}^{+\infty} D(c, \tau)\delta(\tau - t)\mathrm{d}\tau. \qquad (5)$$

The $\delta(t)$ may be replaced with a square wave of $Sa(t)$ as in Fig. 1a, whose height and width are $1/\Delta t$ and $\Delta t$, respectively. As $\Delta t \to 0$, $Sa(t)$ has the same property as that of $\delta(t)$. Replacing $\delta(t)$ with $Sa(t)$, Eq. (5) is,

$$D(c, \tau) = \lim_{\Delta t \to 0} \int_{-\infty}^{+\infty} D(c, t)Sa(t - \tau)\mathrm{d}t. \qquad (6)$$

Similarly, Eq. (4) is

$$V(c, \tau) = \lim_{\Delta t \to 0} \{ \int_{-\infty}^{+\infty} D(c, t)[Sa(t - \tau - \Delta t / 2) - Sa(t - \tau + \Delta t / 2)] \mathrm{d}t \} / \Delta t. \qquad (7)$$

Assuming $V(c, \tau)$ is differentiable, acceleration $A(c, \tau)$ is then,

$$A(c, \tau) = \lim_{\Delta t \to 0} [V(c, \tau + \Delta t / 2) - V(c, \tau - \Delta t / 2)] / \Delta t = \mathrm{d}V(c, \tau) / \mathrm{d}\tau$$
$$= \lim_{\Delta t \to 0} \{ \int_{-\infty}^{+\infty} D(c, t)[Sa(t - \tau - \Delta t) - 2Sa(t - \tau) + Sa(t - \tau + \Delta t)] \mathrm{d}t \} / \Delta t^2 \qquad (8)$$
$$= \mathrm{d}^2 D(c, \tau) / \mathrm{d}\tau^2.$$





### 3.2 Derivatives of non–differentiable position of moving particle

By removing the limiting process, the differentiability of $D(c, t)$ is not required to define $V(c, t)$ and $A(c, t)$. Integrals of Eqs. (5) – (7) are the cross correlation functions between $D(c, t)$ and a set of square waves. These square waves form the observational windows (operators) with which to detect the motion of the particle at time $\tau$. The operator in Eq. (6) detects the position (or displacement) of the particle exposed over interval $\Delta t$, so it is the displacement detector, $Sa(t–\tau) = DDW(t–\tau)$. Similarly the operator in Eq. (7) is the first order difference detector, $Sa(t–\tau–\Delta t/2) – Sa(t–\tau+\Delta t/2) = D1W(t–\tau)$. The operator in Eq. (8) is the second order difference detector, $Sa(t–\tau–\Delta t) – 2Sa(t–\tau) + Sa(t–\tau+\Delta t) = D2W(t–\tau)$. The layouts of these detection windows are in Fig. 1a at time $t = 0$. The $D1W(t)$ and $D2W(t)$ are respectively odd and even functions of time $t$ so that they obey each time reversal property of $d/dt$ and $d^2/dt^2$. We denote $D1W(t–\tau)/\Delta t$ and $D2W(t–\tau)/(\Delta t)^2$ by $VDW(t–\tau)$ and $ADW(t–\tau)$, respectively. The $Sa(t)$ in the definitions may be replaced with other representations for the $\delta(t)$.

### 3. 3 Definition of physical wavelets

The $DDW(t)$ is even, and $VDW(t)$ is odd, and $ADW(t)$ is even with respect to $t$. Therefore $DDW(t–\tau)$ and $VDW(t–\tau)$ are orthogonal to each other at time $t = \tau$, so are $VDW(t–\tau)$ and $ADW(t–\tau)$. But $DDW(t–\tau)$ and $ADW(t–\tau)$ are not. The orthogonality between $DDW(t–\tau)$ and $VDW(t–\tau)$ guarantees that $D(c, \tau)$ and $V(c, \tau)$ are independent of one another. They specify the position of the moving particle in the $D(c, t) – V(c, t)$ plane at time $\tau$, and its motion draws the trajectory in the phase plane. The $A(c, \tau)$ is then uniquely calculated. We name these detection windows "physical wavelets". Physical is prefixed to emphasize that the amplitudes of our wavelets will be fundamental physical quantities like displacement, velocity and acceleration. They become powerful in describing faint precursory signals, buried in chaotic time series, by a displacement–velocity or a displacement–acceleration (force) relationship (Takeda, 1994, 1996). We must find physical laws controlling the precursors to completely prevent the catastrophic events. Deterministic may be prefixed to such precursors.

### 3.4 A brief history of physical wavelets

In 1985, $D1W(t–\tau)$ in Fig. 1b was programed onto a 4–bit LSI of 192 nibble RAM and 2K byte ROM (T8649EBI, TMP47C220AF, 4075, Toshiba Co.) to obtain the speed (velocity) of pressure fluctuations for an oscillometeric digital blood pressure unit design (Takeda, 1993, 1995). The velocity information is imperative for the unit design. If the $D1W(t)$ of Fig. 1a is inverted with a minor scaling adjustment, it is identified as the well–known Haar wavelet. Widening the width of $DDW(t)$, we have the scaling function for the Haar wavelet. The pair becomes a basis of complete orthonormal set, which may offer the simplest multiresolution analysis to time series data (Daubechies, 1992). The formulations of wavelets as in Figs. 1a and 1b are fundamentally different from those in the wavelet analysis. Physical wavelets have been applied to extracting faint deterministic precursors from noisy non–derivative signals to prevent catastrophes of physical systems. They have also been used for studying nonlinear dynamics of time series observed in physical systems, which replace the state spaces, reconstructed by so–called time delay embedding, with physically more tractable phase spaces (Takeda, 1994, 1996;





Takeda and Okada, 2001). Their real–time applications have been privately used in various plants to prevent catastrophes of industrial systems (Takeda, 1994; Takeda et al., 2000). The present concise descriptions are evolved from (Takeda, 2015).

### 3. 5 Equations of EQ motion observed with physical wavelets

We are now able to extract any faint deterministic change embedded in the highly irregular particle position $d(c, m)$ in the EQ time series $\{c\}$. We have the $c$ – coordinate of the particle position smoothed over $\Delta t$,

$$D(c,\tau) = \int_{-\infty}^{+\infty} \{c\}\, Sa(t-\tau)\mathrm{d}t = \int_{-\infty}^{+\infty} \{c\}\, DDW(t-\tau)\mathrm{d}t. \qquad (9)$$

Let the width of $DDW(t)$ be $\Delta t = 2w+1$ events for which $w \geq 1$. Equation (9) is then the moving average of each series of $\{c\}$,

$$D(c,\tau) = [1/(2w+1)] \sum_{j=-w}^{w} d(c,\tau+j). \qquad (10)$$

The interval to take differences in $D1W(t–\tau)$ and $D2W(t–\tau)$ is $\Delta t = 2w+1$. Any interval may be taken for them. Let the interval be an integer of $2n$ for $D1W(t–\tau)$ and another integer $s$ for $D2W(t–\tau)$. Their layouts are in Fig. 1b. If $s = 2n$, the physical wavelets find $D(c, \tau)$ given by Eq. (10), $V(c, \tau)$ and $A(c, \tau)$ by Eqs. (11) and (12),

$$V(c,\tau) = \int_{-\infty}^{+\infty} \{c\}\, VDW(t-\tau)\mathrm{d}t = [D(c,\tau+s/2) - D(c,\tau-s/2)]/s \qquad (11)$$

and

$$A(c,\tau) = \int_{-\infty}^{+\infty} \{c\}\, ADW(t-\tau)\mathrm{d}t = [D(c,\tau+s) - 2D(c,\tau) + D(c,\tau-s)]/s^2. \qquad (12)$$

Equations (11) and (12) are the equations of EQ motion carrying the periodically fluctuating components embedded in $\{c\}$.

### 3. 6 Periodic fluctuations of EQ motion extracted with physical wavelets

The extraction of specific fluctuations in EQ motion is significant if the mutual correlation between the wavelets and $\{c\}$ is high. In the fluctuation (frequency) domain, the extracting function is expressed with the Fourier transform of physical wavelets by the correlation theorem. The respective Fourier transforms of $DDW(t)$, $VDW(t)$ and $ADW(t)$ are then;

$$DDW(f) = \frac{\sin(\pi f \Delta t)}{\pi f \Delta t}, \qquad (13)$$

$$VDW(f) = \frac{2}{\mathrm{i}} \frac{\sin(\pi f \Delta t)}{\pi f \Delta t^2} \sin(\pi f s) \qquad (14)$$

and

$$ADW(f) = -4 \frac{\sin(\pi f \Delta t)}{\pi f \Delta t^3} \sin^2(\pi f s). \qquad (15)$$





Frequency $f$ is in 1/event and $\Delta t = 2w+1$. The symbol i in $VDW(f)$ is a complex number, $i^2 = -1$. The $DDW(f)$ is a low pass filter and $VDW(f)$ is a band pass filter. $ADW(f)$ is another band pass filter whose cut–off frequencies are respectively $(4s)^{-1}$ and $(4s/3)^{-1}$ for high–pass and low–pass filters. They are defined at half the maximum intensity of $ADW(f) \approx (2s)^{-1}$. A functional alternative to $Sa(t)$ improves these filtering functions.

### 5    3.7 Phase plane analyses

The positions of GPS stations are available from the Geospatial Information Authority of Japan (GSI) (GSI, 2017). Denoting geological axis $c$ by $E$ (west–east), $N$ (south–north) and $h$ (down–up), the daily GPS positions are expressed by Eq. (1) where $d(c, m)$ is the $c$–coordinate of a position at day $m$. The daily position of $d(c, m)$ discontinuously changes its direction and speed so that its derivative with respect to day $m$ cannot be defined. We use the physical wavelets to define $D(c, t)$, $V(c, t)$

and $A(c, t)$ and draw the displacement trajectories of daily positions of GPS stations on $D(c, t) – V(c, t)$ plane for which $A(c, t)$ is uniquely defined. We have extracted the anomalous motion of the Pacific Plate and crustal bulges precursory to the 11 March 2011 Tohoku M9 EQ as follows (Takeda, 2011a, 2013, 2015; TEC21, 2017a).

By setting $w = 7$ days and $s = 20$ days, we find a GPS station in Chichijima (in the Pacific), which terminated its operation on 8 Mar 2011, moving westward at a constant speed of $V(E, t) = – 0.23$ mm/day. It has the 28 day periodic modulation

caused by Earth tide of the moon's monthly rotation. The modulation appears as an eastward protruding by the amount of $V(E, t) = 0.15$ mm/day from the constant westward motion of the plate. The protruding width on the $D(E, t) – V(E, t)$ trajectory is 1.5 mm. The anomalous westward motion started on 11 July 2010. On 22 Dec 2010, it reached the largest westward speed of $V(E, t) = – 0.65$ mm/day 4 times larger than the normal speed. On the same day, the M7.9 EQ occurred at the epicenter in the Pacific about 668 km away from the GPS station to the east. The EQ had a normal faulting (JMA, 2017)

in harmony with the westward accelerated motion of the Pacific plate. The westward motion is then rapidly decelerated until it reached $V(E, t) = 0$ mm/day on 4 Feb 2011 (day $t$ in date). Since day $t = m – (s/2 + w) = m –17$ as in Fig. 1b, day $m$ in date is 21 Feb 2011. The motion then turned around eastward and it reached $V(E, t) = + 0.06$ mm/day on 19 Feb 2011 for which day $m$ takes date 8 Mar 2011. It moved the Pacific plate by 1.6 mm eastward 3 days before the 11 March 2011 M9 event. After 22 Dec 2010, the eastward protruding modulation of Earth tide on the $D(E, t) – V(E, t)$ trajectory disappeared. The $D(E,$

$t) – A(E, t)$ trajectory analysis is also available.

Similarly the precursory crustal bulge of the Tohoku area started 16 months before the Tohoku M9 event occurred. The bulge was formed with the horizontal contraction and vertical extension of the crust along the east and west coasts at the plate boundary.  The background noise levels on the daily vertical displacements ($h$ = down–up) are about $\pm$ 20 mm a day. The trajectory drawn on $D(h, t) – V(h, t)$ plane with $w = 200$ days and $2n = s = 300$ days, has the resolutions of 0.1 mm and

0.0001 mm/day 5 orders of magnitude greater than the daily background noise level.  For example, the trajectory of a GPS station at the east coast has linearly changed from – (downward) 0.0046 mm/day to + (upward) 0.0008 mm/day and 1.5 mm lift over 16 months (Takeda, 2011a, 2013, 2015).

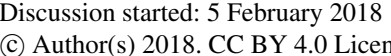



### 3.8 Automatic detection of anomalies leading to the catastrophes of physical systems

The automated real–time detection of anomalies in industrial systems, uses the cross correlation of $V(c, \tau)$ and $A(c, \tau)$, which is proportional to the rate change of the kinetic energy of motion. The rate change is the power that also works as an adaptive filter by setting parameters $w$ and $s$. Any anomaly has been detected by comparing the power with a predetermined threshold

level to prevent the sudden material fractures of rotating heavy machinery systems a few hundreds of milliseconds ahead of time (Takeda, 1994; Takeda et al., 2000; TEC21, 2017d). The real time monitoring of the power has completely prevented sudden drill head's fracture before it fractures at a heavy industrial company. The drilling machine has 5 heads to make many holes of 2 meter length and 20 mm diameter for a nuclear power plant heat exchange system. Other 100% preventions of multi drill head's fracture are those in gun drills in producing large suction rolls for paper mills. The power monitoring

has also been used to detect critical friction states of ball bearings of rotational machineries at a large chemical plant. Time series data for all those rotational machinery systems use the time intervals to make one revolution like $INT$ of EQ source parameters. The other extensive monitoring with the power has been used in locating roughness of the order of 10 microns on the surface profile of working hot rolls in order to polish them during a continuous real time operation of rolling mills.

The same automated detection system as those for industrial systems has detected ionospheric anomalies from the noisy

displacements time series of GPS stations near their epicenters, which may have been perturbed by impending large EQ's. The precursory anomalies appear about two weeks before the large events occur (Takeda and Takeo, 2004).

The cross correlation of noisy or chaotic or even random motions (events) with physical wavelets in real–time, is a reliable method of detecting anomalies leading to the catastrophes of physical systems.

### 3.9 Estimating the number of independent variables creating time series

Statistical analyses of chaotic time series with physical wavelets give us some physical intuition on their quantities. For example, Allan variance (two sample variance) is the averaged kinetic energy proportional to $<V(c, \tau)^2>$ for which $V(c, \tau)$ is detected with velocity detecting wavelet $VDW(c, \tau)$ in Fig. 1a (Takeda, 1996; Takeda and Okada, 2001). Displacement detecting wavelet $DDW(t)$ has been used to estimate the number of independent variables or the number of dynamical degrees of freedom creating time series data (Takeda, 2015). The detection algorithm is the same as that to find false nearest

neighbors for estimating the minimum embedding dimension of an attractor constructed by a delay embedding theorem. The use of $DDW(t)$'s may claim that the minimum embedding dimension is the number of independent variables creating time series data.

With $DDW(t)$'s, we find each cumulative (moving) sum of $2s\,d(INT, i)$ and $2s\,d(DEP, i)$ having three independent variables that are suggested to be three principal stress components acting on the Earth's crust. One principal stress is generally

expected to be normal to the surface with the other two stresses acting in a horizontal plane (Zoback and Zoback, 2002). Since each sum is a scalar, it may be assumed to be proportional to the strain energy density stored in the reginal crust (Takeda, 2015; TEC21, 2017b). The moving sum of $2s\,d(INT, i)$ is then normalized as:



$$NCI(m,2s) = \left[ \sum_{i=m-2s+1}^{m} d(INT,i) \right] \Big/ \left\langle d(INT,mx),2s \right\rangle \max. \qquad (16)$$

Similarly the moving sum of 2s $d(DEP, i)$ is normalized as:

$$NCD(m,2s) = \left[ \sum_{i=m-2s+1}^{m} d(DEP,i) \right] \Big/ \left\langle d(DEP,mx),2s \right\rangle \max. \qquad (17)$$

Here $<d(c, mx, 2s>$max with $c = INT$ or $DEP$ is the maximum of moving sum found at time $m = mx$. The $NCI(m, 2s)$ is proportional to seismic activity. If it is large, the activity is quiet. The $NCD(m, 2s)$ is proportional to seismic depth. If it is

large, the seismic activity is deep. Both of $NCI(m, 2s)$ and $NCD(m, 2s)$ increase to their own peaks at about the same time and then they start to rapidly decrease from their peaks until a large event occurs. The rapid release of strain energy density right before the large event is caused by the increase in the background seismicity, which is known as "Accelerated Moment Release (AMR) (Mignan et el., 2007)". The AMR's on $NCI(m, 2s)$ and $NCD(m, 2s)$ have been observed at every large and great events throughout Japan in both small meshes of about 4° × 5° with $MAG \geq$ about 3.5 and a large region of 16°–52° N

and 116°–156° E with $MAG \geq 4$ (Takeda, 2015). The AMR's in the large region generally start a few days before a large event occurs somewhere in the region as well as before a large aftershock occurs (Takeda, 2015).

## 4 Prediction of impending large events

As an example of many successful test and real predictions by using hypocenter catalogue data (Takeda and Takeo, 2004; Takeda, 2015), we detail a test prediction of the properties of the 1995 Kobe M7.2 event made two months before it occurred.

### 4.1 Anomalous accelerations CQK and CQT precursory to large events

The variations in the stress state of the earth crust make the particle move in an unpredictable way in the EQ space. However, as the shear stress approaches a critical value of a large fault failure, the motion shows two kinds of anomalous accelerations (Takeda and Takeo, 2004, 2007; Takeda, 2011b, 2015; TEC21, 2017c). One is named after the 1995 Kobe M 7.2 as CQK. The CQ and K stand for Critical Quiescence and Kobe, respectively. Another is named after the 2000 Tottori M 7.2 as CQT

and the T stands for Tottori. The CQK or CQT is found precursory to almost every large EQ and swarm (M > about 6) throughout Japan including the 2011 great Tohoku M9 (Takeda, 2015). There are a few exceptions preceded by medium size swarms, which is detailed in (Takeda, 2015).

To show the CQK and CQT, we have a small mesh region of $LAT = 32°–36°$ N and $LON = 131.5°–136.5°$ E as in Fig. 2a. The regional area lies along the tectonic plate boundary between the subducting Philippine Sea Plate and the southern edge

of the Eurasian Plate (the Amurian Plate). The subducting slab of the Pacific Plate is under these plates. The Philippine Sea Plate is moving to the northwest at about 3cm per year with respect to the Eurasian Plate, which accumulates the shear stress





of about 0.01 MPa per year in this region (Takeda, 2015). It suggests that the stress accumulation creates the shallow EQ events in Figs. 2b and 2c.

The shallow particle motion, during the period of 1986–2000, is shown in Fig. 3a for which the EQ events with $MAG \geq 3.5$ and $DEP \leq 300$ km are collected from the Japan Metrological Agency (JMA) hypocenter catalogues of 1983–1997 (JMA,
2017) and JMA unified hypocenter catalogues of 1997–2012 (JMA, 2017; NIED, 2017). We assume that this selection from the regional EQ catalogues is complete and makes the particle move in a closed physical system. These assumptions appear to be valid except for the motion near the regional boundary (Takeda, 2015).

The $d(c, m)$ is highly irregular as in Fig. 3a; however, its time series $\{c\}$ has the spectral peaks common to all EQ source parameter $c$, which are neither those of resonances characterizing linear systems nor artifacts as in Fig. 4 (Takeda, 2011b,
2015). One of the peaks is at about 64 events (approximately 600 days) in this region. Only the periodicity within about 60–70 events, which are embedded in $\{c\}$, can give us CQK and CQT for this region. We extract this periodic oscillation by setting $w = 12$ and $s = 35$ in Eqs. (10) and (12). The periodicity in every other mesh region (about $5^\circ \times 5^\circ$) throughout Japan is about 40–70 events, which is extracted by setting $w = 15$–25 events and $s = 20$–35 events (Takeda, 2015). The EQ's are collected with $MAG \geq 3.5$ (or $MAG \geq 3.3$ in a few areas) and all $DEP$.

The anomalous acceleration is labelled as CQK or CQT at the downward red arrow on the pair of blue $A(INT, \tau)$ and black $A(DEP, \tau)$ in row $INT$ of Figs. 3a and 3b. They constitute the phase inversion between $A(DEP, \tau)$ and $A(INT, \tau)$ with the negative amplitude of $A(MAG, \tau)$ (Takeda and Takeo, 2004, 2007; Takeda, 2011b, 2015; TEC21, 2017e). After the phase inversion, the 1995 Kobe M 7.2 and the 2000 Tottori M 7.2 occurred at the next negative and positive peaks of $A(INT, \tau)$ (at the short arrows in row $INT$), respectively. The phase inversions of displacement $D(c, t)$ at CQK and CQT are referred to as
CQKD and CQTD, respectively (Takeda, 2015). The CQKD for the 1995 Kobe M 7.2 is shown in Fig. 5.

### 4.2 Equation of EQ periodic motion

The periodic fluctuations of $D(c, \tau)$ and $A(c, \tau)$ may be expressed by the restoring force $F(c, \tau)$ as $F(c, \tau) \approx A(c, \tau) \approx -K(c) \times D(c, \tau)$ with a positive constant $K(c)$ which is a weak function of time $\tau$. Hereafter, the $D(c, \tau)$ is the displacement of the periodic oscillation so that each origin is not at its own graphical reference. The origin is also a weak function of time.

Define the unique time during CQK or CQT as follows. The time at which $A(c, \tau)$ takes its first negative (or positive) peak amplitude is $\tau a$. The time at which $A(c, \tau)$ becomes zero after the first peak is $\tau b$. The time at which $A(c, \tau)$ takes the second peak is $\tau r$. Parameter $c$ may establish its own $\tau a$, $\tau b$ and $\tau r$. The impending large event ruptures at time $\tau r$ specified for $c = INT$ and $DEP$ as well (Takeda, 2015).

### 4.3 Prediction of fault size, motion, and its magnitude

The vertical component of the particle motion in the earth crust finds the absolute magnitude of displacement, $|D(DEP, \tau a)|$, during CQK or CQT to be comparable to the planar fault width $W$ of the impending large events throughout Japan (Takeda, 2015). This suggests $F(DEP, \tau a)$ has induced the shear stress that exceeds the critical failure value of a local fault plane of





width $W$ (Takeda and Takeo, 2010; Takeda, 2015). The $F(DEP, \tau a)$ points to the shallower (up) and the deeper (down) depth for CQK and CQT, respectively. Similarly the horizontal restoring force $F(c, \tau a)$, where $c = LAT$ and $LON$, may induce shear stress on the planar fault as $F(DEP, \tau a)$ does. The total length of the horizontal displacements $D(c, \tau a)$ in degrees is then expected to be comparable to the fault length $L$ in km as $|D(DEP, \tau a)| \approx W$.

As for the 1995 Kobe M7.2 CQK event, the observation finds; $|D(DEP, \tau a)| = 20$ km, $|D(LAT, \tau a)| = |-0.25°|$ and $|D(LON, \tau a)| = |-0.53°|$ in Figs. 3a, 3b and 6. The total length is then 56.4 km. These predictable $W$ and $L$ nearly match with the seismological observation of $W = 20$ km and $L = 40$ km (Kikuchi and Kanamori, 1996). The $F(LAT, \tau a)$ and $F(LON, \tau a)$ point to $(LAT, LON) = (0.25°, 0.53°)$ so that the net force direction is 60° from the north to the east. It suggests that the force has induced the shear stress by which the hanging wall block will be moved relative to the footwall block along the fault line

toward the net force direction. The fault motion is then right–lateral strike–slip.

This predictable motion is in good harmony with that suggested by the focal mechanism of (Strike, Dip, Rake) = (233°, 86°, 167°) (Kikuchi and Kanamori, 1996; JMA, 2017). The motion is right–lateral strike–slip with slightly upward dip–slip component. The shear stress points to 53° which is nearly the predictable direction of 60°. This directivity of more than 45° is suggested by the magnitude relation in $A(LON, \tau) > A(LAT, \tau) > 0$ indicated by arrow at CQK in Fig. 3b.

Deterministic prediction of the EQ magnitude is made with the prediction of the planar fault length $L$ and width $W$ in km. The EQ magnitude is estimated by an empirical law of $M = \log S + 3.9$ where $S = L \times W$ each of which is estimated by the aftershock distribution (Utsu, 2002). As for the Kobe CQK event, the $M$ is 6.9. The seismological observation of $M$ is 6.9 and 7.2 for the moment and the JMA magnitude, respectively. The estimation of JMA $M$ with an assumption of $L = 2W$ (predicted $W$) for most of the large events throughout Japan are also in good agreement with the observation of $M$ (Takeda,

20    2015).

### 4.4 Rupture time prediction

The rupture time $\tau r$ in $m = \tau r + s + w$, can be predicted within one or two event time accuracy by using $D(INT, \tau r)$ and $A(INT, \tau r)$ (Takeda and Takeo, 2004, 2007; Takeda, 2011b, 2015). For example, the Kobe EQ of January 17 1995 has the test prediction made on October 24 1994 with 19 events ahead of time (*Takeda*, 2015). The Kobe EQ ruptured in 19 events as in

Fig. 6. The conversion from the event time to real time requires an average rate of events (Takeda and Takeo, 2004, 2007; Takeda, 2011b, 2015). There are considerable time delays from $\tau r$ for some CQK events. These CQK events all have very similar focal mechanisms that show a significant up–ward dip–slip component in them. The delay suggests that the first upward stress loading is not strong enough to exceed the critical value of the fault failure. Thus another upward loading may have been required, which is predictable (Takeda, 2015).

### 4.5 Epicenter prediction


As for the epicenter prediction, the $D(c, t)$ with $c = LAT$, $LON$ and $DEP$ during CQK and CQT, may be linearly extrapolated to the predicted rupture time $\tau r$ (Takeda and Takeo, 2004, 2007; Takeda, 2011b, 2015). For example, the test prediction of





the Kobe EQ is (34.53º, 135.18º, 20.8km) in (*LAT*, *LON*, *DEP*) as in Fig. 6, which matches with the seismological observation of (34.595º, 135.038º, 16.06 km) in JMA hypocenter catalogue.

### 4.6 Rupture time and epicenter area predictions by monitoring strain energy density time series

Locating of AMR on the normalized strain energy density of $NCI(m, 2s)$ and $NCD(m, 2s)$, defined by Eqs. (16) and (17), is
incorporated into deterministic predictions (Takeda, 2015). For example, by setting $2s = 70$ for the events of $MAG \geq 3.5$ and $DEP \leq 300$ km in the same region of 32º–36º N and 131.5º–136.5º E as that in Fig. 2a, the AMR on the 1995 Kobe M 7.2 is located as in Fig. 7a. The peaks of $NCI(m, 70)$ and $NCD(m, 70)$ are both at about $m = 556$ (6 Apr 1994) with a quiet and deep seismicity. During the rapid release of strain energy into active shallower seismicity, the 1995 Kobe event occurred at $m = 602$ (17 Jan 1995). Other AMR's on the 1997 Yamaguchi M6.7 and the 2000 Tottori M7.2 give some variations on the
predictive date by their AMR monitoring. The prediction accuracy can be improved by selecting different $MAG$ and $2s$. The region may be selectively chosen smaller than the 4°×5° to narrow the region to the epicenter area. The AMR's in large regions generally have their peaks a few days before a large event occurs somewhere in the region. For example, the AMR of the 1995 Kobe M7.2 in the large region of 16°–52° N and 116°–156° E, is shown in Fig. 7b. The $NCI(m, 30)$ has a peak at $m = 9892$ (14 Jan 1995 at 04:49); whereas, $NCD(m, 30)$ has a peak at $m = 9906$ (16 Jan 1995 at 08:53). The Kobe event
occurred at $m = 9909$ (17 Jan 2015 at 05:40).

The AMR's on a sequence of a precursory event, fore event, and main event, may also be used for their date and time predictions as those on the 2011 Tohoku M9. The sequence appears as a cycle of strain energy accumulation and release to the M9 event, which is shown in Figs. 8a and 8b. As for the M9 event, our website (www.tec21.jp) had posted the $NCI(m, 2s)$ and $NCD(m, 2s)$ on 4 Mar 2011 before the 11 Mar 2011 M9 event. The strain energy cycle has often been updated at
www.tec21.jp since 2003 before large events occurred.

### 5. Seismogenic origin of periodicities

### 5.1. Periodic fluctuations in EQ event time series

The periodic fluctuation embedded in time series $\{c\}$ is from the two time dependent unique seismogenic structures. Each structure, which is the hypothesized B–D coupling, generates its own EQ events characterized with magnitude $Mc$ (Jin and
Aki, 1989; Aki, 1996). The structure in a small seismic region of 120 km radius and its $Mc$ are systematically measured by the temporal variation of the decay rate of seismic coda waves (coda $Q^{-1}$). If coda $Q^{-1}$ is fast (large $Q^{-1}$) in a region, its regional $Mc$ is from 4 to 4.5. If coda $Q^{-1}$ is slow (small $Q^{-1}$) in a region, its regional $Mc$ is from 3 to 3.5. The coda $Q^{-1}$ has positive simultaneous correlation with the respective seismicity of $Mc$ over 50 years.

As for the $Mc$ time series analysis, the seismicity of $Mc$, denoted by $N(Mc)$, is defined as the ratio of $Mc$ events to the total
events of $MAG \geq 3.0$ occurred during the time to count 100 successive $Mc$ events. The observational window of 100 events



has 25 events overlapped for the next observation. The time axis of $N(Mc)$ series is real time with which each series shows the periodicity of about 10 years.

The periodicity of $N(Mc)$ in the region of 120 km radius may be translated into that of $D(INT, t)$ in the 4.6 times larger region of Fig. 2a. Making the number of moving average of $D(INT, t)$ increase from 25 events ($w = 12$) to 100 events, we have the

same observational time window as that to obtain $N(Mc)$. The periodicity of $D(INT, t)$ with $MAG \geq 3.5$ ($3.5 \approx Mc$) is then increased from about 2 years (600 days) to about 8 years (Takeda, 2011b, 2015). This increase is because the smaller periodic fluctuations are all filtered out by the filtering function in Eq. (13). Since the most of seismic events of $MAG \geq 3.5$ is those around $Mc$, $D(c, t)$ is also dominated by the seismicity of $Mc$ ($3 \leq Mc \leq 3.5$ or $4 \leq Mc \leq 4.5$). The periodicity of $D(c, t)$ is in harmony with that of $N(Mc)$. It suggests that periodic fluctuations (the scale dependent EQ phenomena) are embedded

in time series {c} of seismic events.

### 5.2 Stress loading at CQK and CQT

Each positive correlation of the coda $Q^{-1}$ with the respective seismicity of $Mc$ is destroyed about a year and half ahead of large events (Aki, 2004; Jin et al., 2004) with the increased seismicity of magnitude of about 4 as those in CQKD and CQTD. Thus it also suggests we are observing the same anomaly in different observational windows.

The coda $Q^{-1}$ map of Japan (its frequency within 1–4 Hz) shows many local spots distinctively separated by large and small $Q^{-1}$ (Jin and Aki, 2005). The CQK's are all in the small $Q^{-1}$ spots, whereas the CQT's are in the large $Q^{-1}$ spots throughout Japan (Takeda, 2015). This suggests the two anomalies on particle motions in the EQ space, CQK and CQT, may represent the two unique seismogenic structures each of which induces upward (compressional) and downward (dilatational) stress loading to the fault zones, respectively. These stress loadings couple with the regional stress accumulation, which appear to

create various fault mechanisms.

### 6 Conclusions

Physical wavelets find the periodic equations of EQ motions from the non–derivative (chaotic) time series of EQ source parameters $c$. They are $F(c, \tau) \approx A(c, \tau) \approx - K(c) \times D(c, \tau)$ where $F(c, \tau)$, $A(c, \tau)$, $K(c)$ and $D(c, \tau)$ are restoring force, acceleration, a positive constant $K(c)$ and displacement, respectively. $K(c)$ is a week function of time $\tau$. The parameters are $c$

$= LAT$ (latitude), $LON$ (longitude), $DEP$ (focal depth), $INT$ (inter event time), and $MAG$ (magnitude). Acceleration $A(c, \tau)$ among $c = INT$, $DEP$ and $MAG$ shows the phase inversion between $A(DEP, \tau)$ and $A(INT, \tau)$ with the negative amplitude of $A(MAG, \tau)$ weeks and months before the large EQ's ($MAG \geq$ about 6) occurred throughout Japan (Takeda, 2015). The precursory periodic motion gives us the deterministic prediction on the fault size and motion, the rupture time, and the epicenter of impending large and great EQ's. Various prediction algorithms, which are based on physics of the periodic

motion described in this article, are detailed in (Takeda, 2015) to make automated predictions possible for any tectonically active regions that have a seismic network detecting EQ's of $M \geq$ about 3. The prediction system (Takeda, 2015), which is





yet to be fully operated, extensively uses the AMR on the strain energy density time series in selective regions and the GPS geological displacements analyses by using physical wavelets. The wavelets are capable of extracting deterministic precursors embedded in the highly irregular time series of physical systems. They may also formulate physical laws for precursory phenomena of natural and earth systems to mitigate their hazards.

**Acknowledgements**

The EQ source parameters used are from JMA hypocenter catalogs and JMA unified hypocenter catalogs whose websites are listed in the references.

This study had been inspired and mentored by four late scientists. They are a seismologist, Keiiti Aki (3 March 1930 – 17 May 2005), a physicist and my dissertation advisor, Makoto Takeo (6 April 1920 – 23 May 2010), a physicist, Gertrude

Rempfer (30 January 1912 – 4 October 2011) and a fluid engineer and my father, Rikiya Takeda (27 September 1923 – 7 March 2011).

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



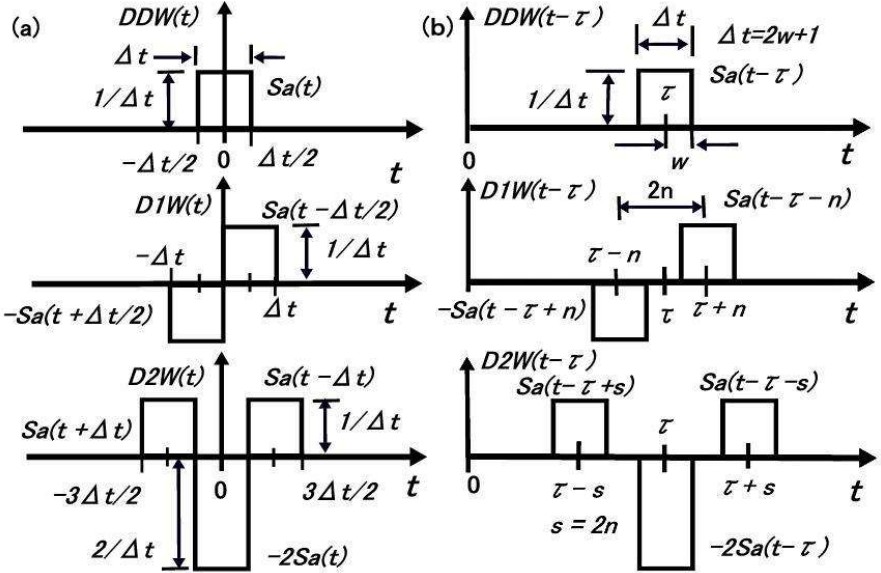

**Figure 1.** The layouts of square waves to construct physical wavelets. (a) The interval to take differences in $D1W(t)$ and $D2W(t)$ is $\Delta t = 2w+1$ that is the width of $Sa(t)$ or $DDW(t)$. (b) The interval can be any integer different from width $\Delta t$ of

5   $Sa(t–\tau)$. The layouts are shown for $D1W(t–\tau)$ and $D2W(t–\tau)$ with $s = 2n$ where $s > \Delta t$. A few other possible layouts with different conditions among $s$, $2n$ and $\Delta t$ are not shown.





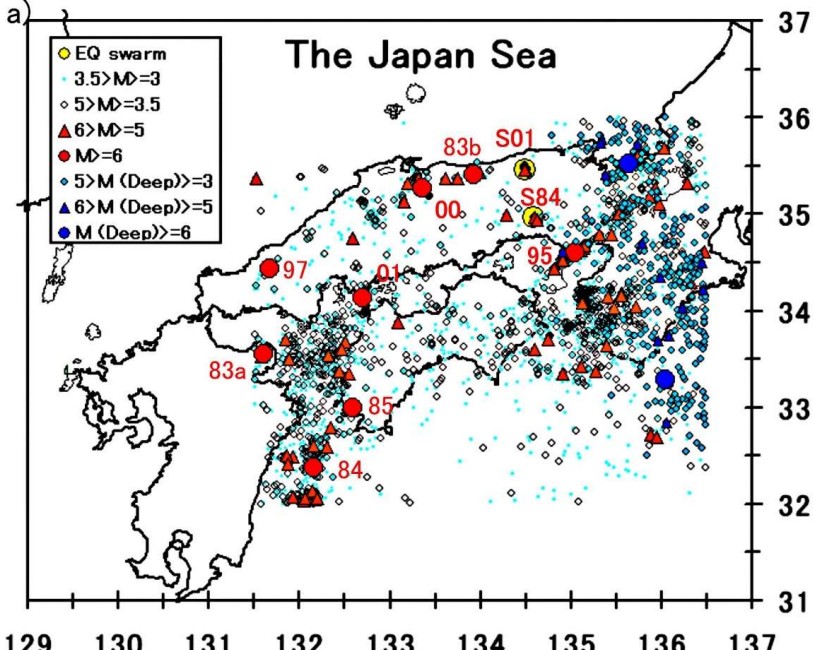



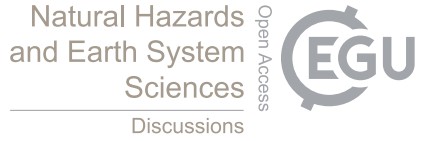

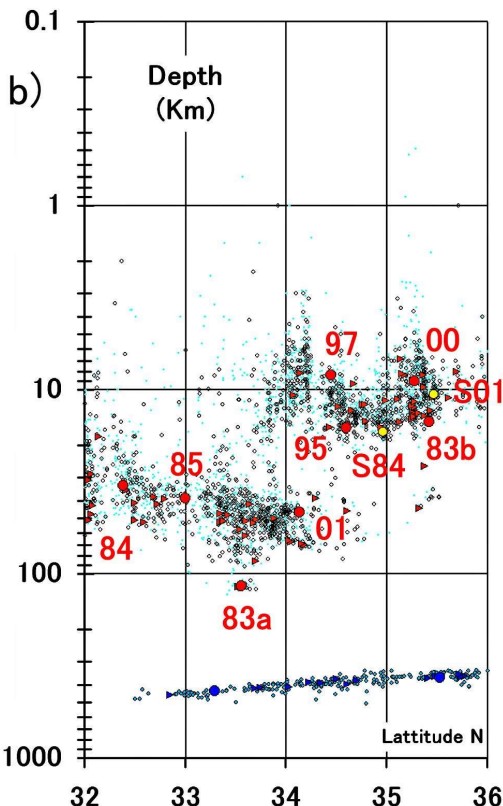

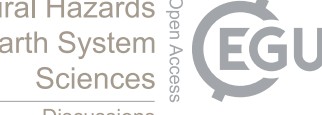



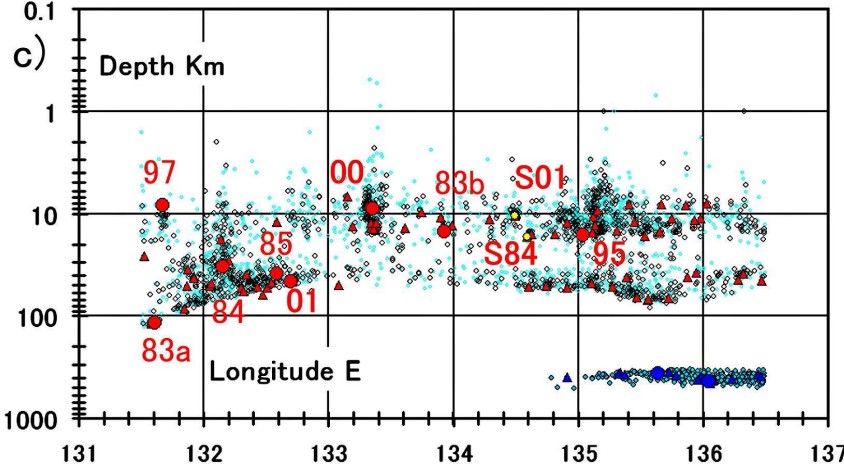

**Figure 2.** (a) Seismicity in a small mesh region of *LAT* = 32°–36° N and *LON* = 131.5°–136.5° E. (b) Latitudinal cross sectional view. (c) Longitudinal cross sectional view. The regional area lies along the tectonic plate boundary between the subducting Philippine Sea Plate and the southern edge of the Eurasian Plate. The subducting slab of the Pacific Plate is under these plates. The EQ events of *MAG* ≥ 3 are collected from the Japan Metrological Agency (JMA) hypocenter catalogues of 1983–1997 and JMA unified hypocenter catalogues 1997–2012. The shallow events are in and above the subduction zone at *DEP* ≈ 30–100 km. The watercolor dots are the EQ's of 3 ≤ *MAG* < 3.5. Black circles are the EQ's of *MAG* ≥3.5. The yellow dots are two EQ swarms, labelled as S84 and S01 (the 1984 and 2001 swarm). The red triangles are the EQ's of 5 ≤ *MAG* < 6. The red circular dots are the large EQ's of *MAG* ≥ 6 whose labels are the event years. The 83b is the 1983 Misasa *M* 6.3. The 95, 97, and 00 are the 1995 Kobe M7.2, the 1997 Yamaguchi M6.7, and the 2000 Tottori M7.2, respectively. The deep EQ events are along the subducting slab of the Pacific Plate. Sky blue dots enclosed with a black circle are for deep EQ's of 3 ≤ *MAG* < 5. The blue filled triangles and circles are the deep EQ's of 5 ≤ *MAG* < 6 and *MAG* ≥ 6, respectively.

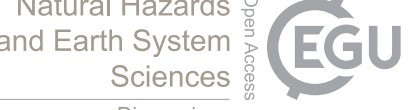



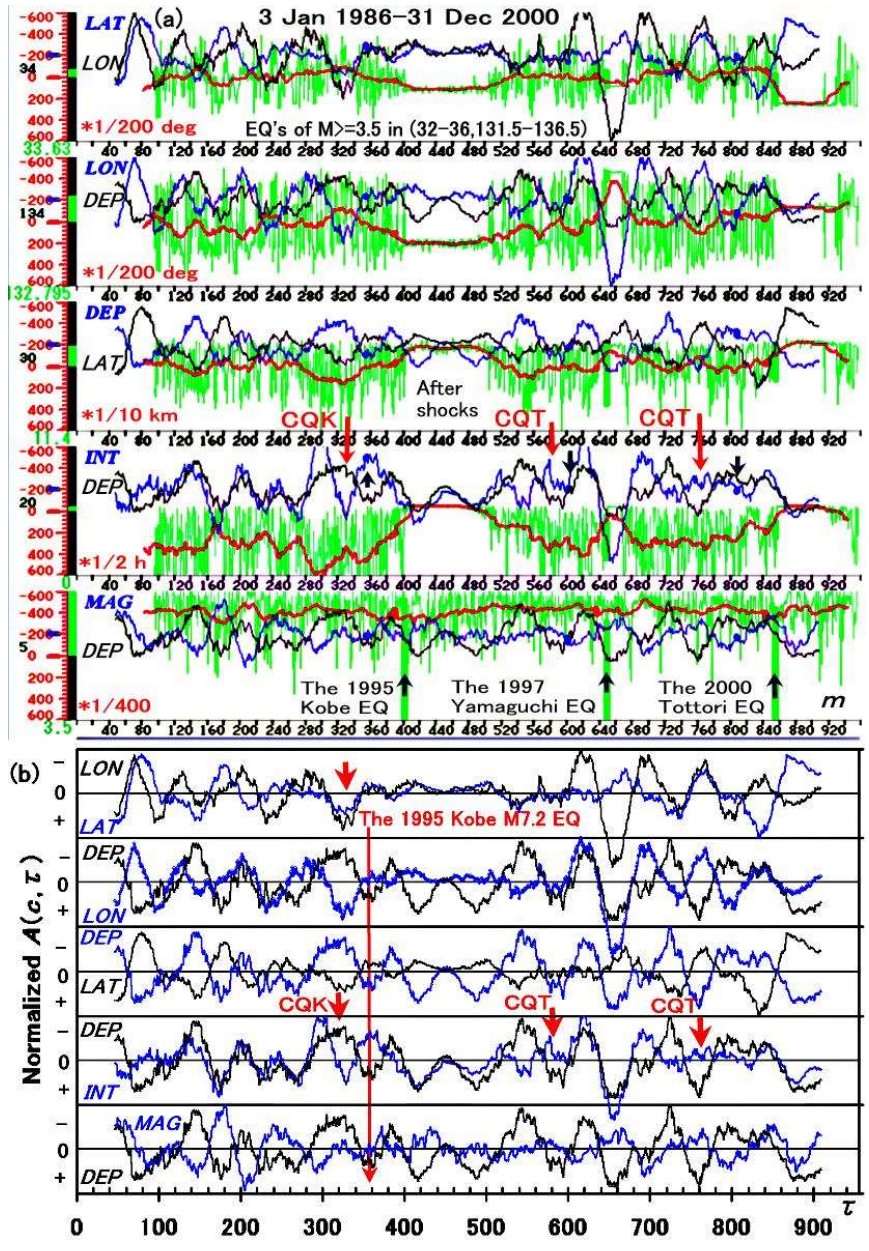





**Figure 3.** (a) The shallow particle motion in the EQ space whose *c* axis is the vertical axis in row *c*. All the horizontal scales are in chronological event index *m* which shares other time *t* and *τ* where *m* = *t* + *w* and *t* = *τ* + *s* (*w* = 12 and *s* = 35). The date at *m* = 0 and 1 is 3 Jan 1986 and 5 Jan 1986, respectively. The *d*(*c*, *m*) in green is the relative position from each graphical reference of 34° N, 133° E, 30 km, 20 hours and 5. They are all at zeroes of the manometer like scales on the left.

The scale is magnified 200 times for *LAT* and *LON*, 10 times for *DEP*, 2 times for *INT*, and 400 times for *MAG*. For example, the *LAT*, *LON*, *DEP*, and *INT* at scale – 200 corresponds to 33° N, 132° E, 10 km, and 100 hours, respectively. The *MAG* range is from 3.5 (–600) to 6.5 (600) so that it saturates above 6.5. Each manometer column shows a variation of *d*(*c*, *m*) with its digital reading during monitoring. The last readings at *m* = 956 (31 Dec 2000) are *LAT* = 33.63° N, *LON* = 132.795° E, *DEP* = 11.4 km, *INT* = 0 (0.2165) hours, and *MAG* = 3.5. The positive direction is downward with respect to each

reference point. The relative position *D*(*c*, *t*) and acceleration *A*(*c*, *τ*) show the periodic fluctuations of about 70 (2*s*) events. The *D*(*c*, *t*) is in red and *A*(*c*, *τ*) in blue and black. The black *A*(*c*, *τ*) is, from the top axis, *c* = *LON*, *DEP*, *LAT*, *DEP*, and *DEP*. Their relative amplitudes are with respect to each origin of blue bar marked at –200 on the left scales. The *d*(*c*, *m*), *D*(*c*, *t*) and *A*(*c*, *τ*) becomes bold at the events of *MAG* ≥ 6. The bold lines of *d*(*MAG*, *m*) with black arrows have the EQ names of the 1995 Kobe M7.2 (at *m* = 402, 17 Jan 1995), the 1997 Yamaguchi M6.7 (at *m* = 649, 25 June 1997) and the 2000 Tottori

M7.2 (at *m* = 856, 6 Oct 2000). The black arrows on *A*(*INT*, *τ*) indicate their temporal locations. (b) Normalized pair of *A*(*c*, *τ*). Each pair of *A*(*c*, *τ*) in blue and black in Fig. 3a is normalized with respect to its own maximum of *A*(*c*, *τ*) as in Fig. 3b. Every amplitude of *A*(*c*, *τ*) > 0 is within + one. Each pair of normalized blue and black *A*(*c*, *τ*) are in the + and – regions as in Fig. 3a. The first and last 47 events for *A*(*c*, *τ*) are not obtained because of *m* = *τ* + *s* + *w* (*w* = 12 and *s* = 35). The location of the 1995 Kobe M7.2 EQ is indicated with the long–downward arrow. The short arrows in *DEP–INT* row are at CQK and

CQT. Blue *A*(*LAT*, *τ*) and black *A*(*LON*, *τ*), pointed by another short arrow above *CQK*, indicates *A*(*LON*, *τ*) > *A*(*LAT*, *τ*) > 0.

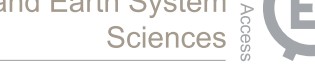



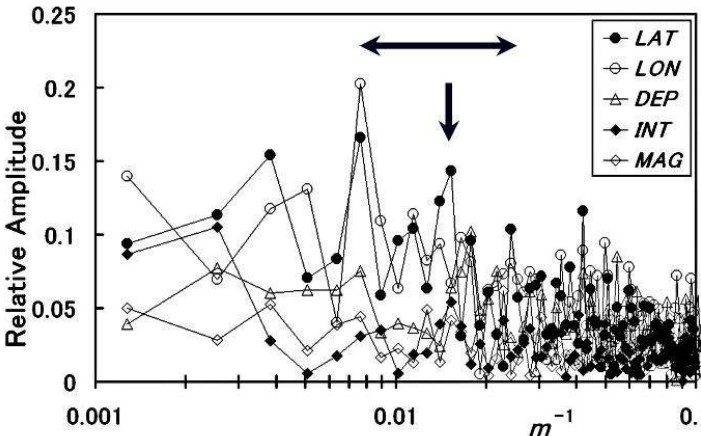

**Figure 4**. The spectra of $d(c, m)$ from 1984 to the 1995 Kobe EQ. The EQ particle motion has larger fluctuation amplitudes in *LON* than in *LAT* as seen in Fig. 3. This is because if the EQ particle in Figs. 2a –2c moves from the Kobe area (labelled as 95) to the Yamaguchi area (labelled as 97), the variation in *LON* is much larger than that in *LAT*. This causes a spectral split of *LON* at the common peak of about 64 events on each spectrum, which is shown with the vertical arrow. The horizontal arrows show the frequency range of extraction, which is the half width of $ADW(f)$ with $w = 12$ and $s = 35$.

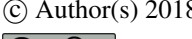



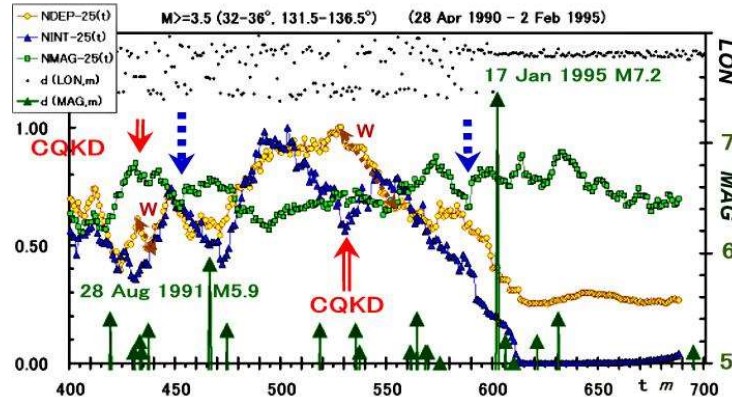

**Figure 5.** The CQKD of $D(DEP, t)$, $D(INT, t)$ and $[D(MAG, t) – 3]$ with $2w+1 = 25$ events in Eq. (9), are normalized to NDEP–25(t), NINT–25(t) and NMAG–25(t) with the past maximum values of 47.16 km, 331.33 h, and 1.4, respectively. At the Kobe CQKD of $t = 528$ (28 June 1993), $D(DEP, t)$, $D(INT, t)$ and $D(MAG, t)$ are 47.16 km, 195.48 h and 4.0, respectively. The *MAG* scales of $d(MAG, m)$ are on the right for which $MAG \geq 5$ is shown in up arrows. The $d(LON, m)$ is relative on the right. The 1995 Kobe event in dot arrow is at $t = 590$ ($m = 602$).



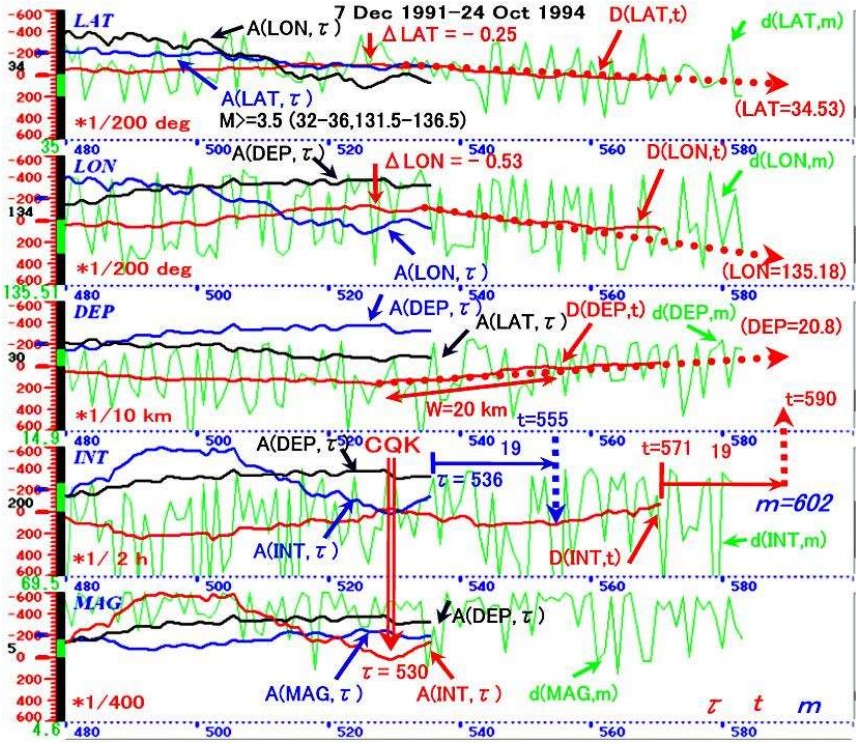

**Figure 6.** A prediction test on the 17 Jan 1995 Kobe EQ is made on 21 Oct 1994 at time $\tau = 536$. In row *INT*, we locate the rupture time $\tau r$ to be in 19 events at $t = 555$ on a dip of $D(INT, t)$. The linear extrapolations of $D(c, t)$ in dot arrows are the predicted epicenter location in $(LAT, LON, DEP) = (34.53, 135.18, 20.8)$. The predicted fault width $W$ along on $D(DEP, t)$ is 20 km.





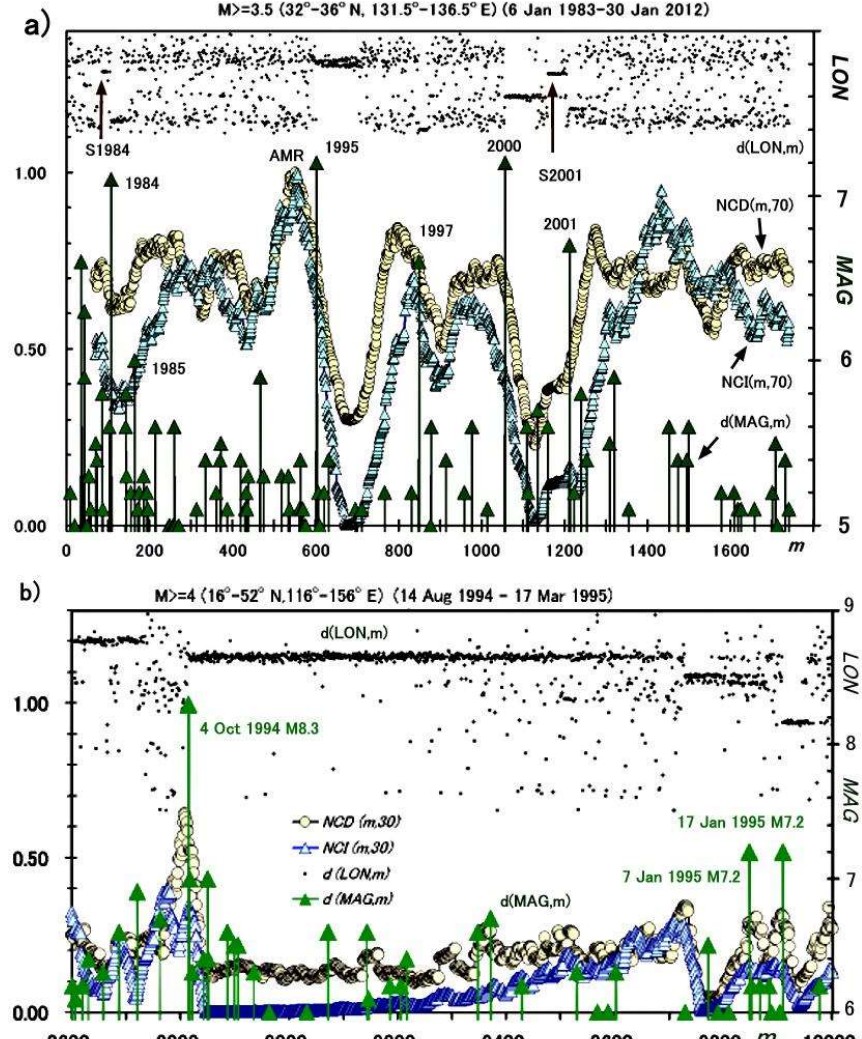

**Figure 7. (a)** The normalized strain energy density time series $NCI(m, 70)$ and $NCD(m, 70)$ from 6 Jan 1983 to 30 Jan 2012 in the small region of $LAT = 32°–36°$ N and $LON = 131.5°–136.5°$ E as in Fig. 2a. Their time series are made from shallow

5  EQ's of $MAG \geq 3.5$ and $DEP \leq 300$ km as in Figs. 2a, 2b and 2c. The large EQ's are labelled on $d(MAG, m)$ in full year. For example, two EQ swarms, labelled as S84 and S01 in Fig. 2a are S1984 and S2001, respectively. The figure axes are the same as those of Fig.5. The AMR on the 1995 Kobe M7.2 is labelled on $NCI(m, 70)$ and $NCD(m, 70)$. Time $m$ has the



following corresponding date: $m$ = 200 to 3 Jan 1986; $m$ = 400 to 28 Apr 1990; $m$ = 600 to 16 Jan 1995; $m$ = 800 to 26 Jun 1996; $m$ = 1000 to 28 Dec 1999; $m$ = 1200 to 8 Feb 2001; $m$ = 1400 to 27 Oct 2004; $m$ = 1600 to 15 Jul 2009.

(b) The $NCI(m, 30)$ and $NCD(m, 30)$ from 14 Aug 1994 to 17 Mar 1995 in the large region of $LAT$ = 16°–52° N and $LON$ = 116°–156° E. Their time series are made from all EQ's of $MAG \geq 4$ from JMA hypocenter catalogues of 1983–1997. The AMR on the 17 Jan 1995 M7.2 (Kobe event) followed those on the 4 Oct 1994 M8.3 and the 7 Jan 1995 M7.2. Time $m$ has the following corresponding date: $m$ = 8800 to 27 Sep 1994; $m$ =9000 to 5 Oct 1994; $m$ =9200 to 9 Oct 1994; $m$ = 9400 to 21 Oct 1994; $m$ = 9600 to 22 Nov 1994; $m$ = 9800 to 30 Dec 1994.

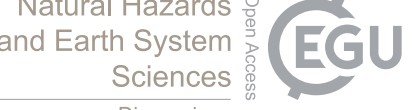

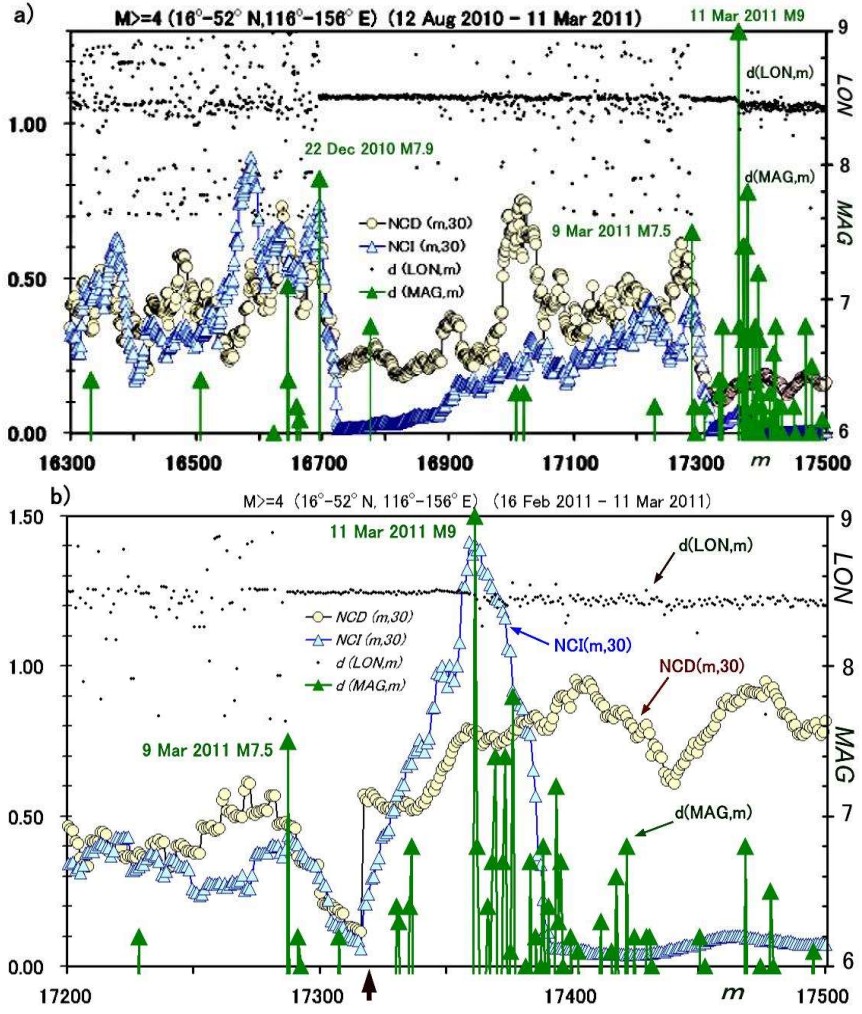

**Figure 8.** (a) The normalized strain energy density time series of $NCI(m, 30)$ and $NCD(m, 30)$ from 12 Aug 2010 to 11 Mar 2011. Their time series are made from all EQ's of $MAG \geq 4$ collected from the JMA unified hypocenter catalogues for the region of $LAT = 16°–52°$ N and $LON = 116°–156°$ E. The figure axes are the same as those of Fig.5 and Figs.7a and 7b. The 22 Dec 2010 M7.9 at $m = 16695$ is a precursory event to the 11 Mar 2011 Tohoku M9, which occurred near Chichijima (an island in the Pacific Ocean). A large foreshock to the M9 event at $m = 17287$ is the 9 Mar 2011 M7.5. Time $m$ has the

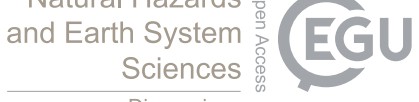



following corresponding date: $m$ = 16500 to 3 Oct 2010; $m$ =16700 to 22 Dec 2010; $m$ =16900 to 28 Dec 2010; $m$ =17000 to 10 Jan 2011; $m$ =17100 to 25 Jan 2011; $m$ =17200 to 16 Feb 2011; $m$ =17300 to 9 Mar 2011.

(b) The AMR's on the normalized and magnified strain energy density time series of $NCI(m, 30)$ and $NCD(m, 30)$ from $m$ = 17200 (16 Feb 2011 at 02:23) to 17500 (11 Mar 2011 at 22:35). The $NCI(m, 30)$ and $NCD(m, 30)$ are respectively magnified

5    by 20 and 5 times after $m$ = 17317 (9 Mar 2011 at 16:56) indicated with the up-arrow on the $m$ axis. The peaks for $NCI(m,$ 30) and $NCD(m, 30)$ are at $m$ = 17359 (11 Mar 2011 at 13:12) and $m$ = 17358 (11 Mar 2011 at 10:41), respectively. The 11 Mar 2011 M9 (the 2011 Tohoku M9) occurred at $m$ = 17361 (11 Mar 2011 at 14:46). Time $m$ has the following corresponding date and time: $m$ = 17300 to 9 Mar 2011 at 13:04; $m$ = 17400 to 11 Mar 2011 at 16:36.