# Peer review of "Physical laws for precursory phenomena of impending large earthquakes and their applications to predictions"

_Natural Hazards and Earth System Sciences, 2017_

## Referee Comment (RC1) · Anonymous Referee #1 · 6 Apr 2018

In this paper, Fumihide Takeda is trying to investigate the problem of earthquake prediction. Particularly, in this study, the author introduces a virtual particle of unit mass in a property space. The coordinates of this space are the latitude, the longitude, the depth, the time interval between consecutive events and the magnitude and the rupture time. The rupture time here plays the role of a chronological event index. Studying the motion of the virtual particle, periodic anomalous accelerations were detected weeks and months before large impending earthquakes. The periodic particle motions are related to the fault size and motion, rupture time, and the epicenter of impending large earthquakes. The interesting approach and the good overall quality of the present study are needed to be supported with more detailed analysis of the existing results.

[Figure]

Specific comments:

(a) The present study for the Prediction of fault size, motion, magnitude and rupture time must be extended for all the cases with a magnitude over a specific threshold for the better evaluation of the suggested method.

(b) The present study must be framed by the appropriate statistical analysis of the results including the false alarm rate.

(c) This analysis is closely related to the natural time analysis in which the order of the event (as an index) is also considered as one of the main characteristics of the examined time series. I am suggesting the following two references to be included:

Natural-time analysis of critical phenomena: The case of seismicity PA Varotsos, NV Sarlis, ES Skordas, S Uyeda, M Kamogawa EPL (Europhysics Letters) 92 (2), 29002

Natural time analysis of critical phenomena P Varotsos, NV Sarlis, ES Skordas, S Uyeda, M Kamogawa Proceedings of the National Academy of Sciences 108 (28), 11361-11364

(d) The analysis of the "Automatic detection of anomalies leading to the catastrophes of physical systems" needs improvement including for example among others a cross-correlation diagram respect to the time.

Technical corrections:

(a) The quality of the figures needs improvement

(b) Grammatical errors:

Page 1, line 26: with Global -> by Global

Page 1, line 30: on crustal surface -> on the crustal surface

Page 2, line 4: As an example of many test -> As an example of many tests

Page 4, line 23: oscillometeric -> oscillometric

Page 4, line 26: complete -> a complete

Page 6, line 26: Similarly -> Similarly,

Page 7, line 31: reginal -> regional

Page 8, line 4: Similarly -> Similarly,

Page 8, line 14: every large and great events -> every large and great event

Page 10, line 2: Similarly -> Similarly,

Page 10, line 14: by arrow -> by an arrow

Page 11, line 16: and main event, -> and main event

Page 11, line 17: The sequence appears as a cycle of strain energy accumulation and release to the M9 event, -> release: wrong grammar

Page 11, line 24: characterized with magnitude -> characterized by magnitude

Page 12, line 12: about a year and half -> about a year and a half

Page 12, line 19: which appear -> which appears

Page 12, line 28: prediction on the fault size and motion -> prediction of the fault size and motion

---

## Referee Comment (RC2) · A. De Santis (Referee) · 6 Apr 2018

Angelo De Santis

This work introduces some functions called by the author "physical wavelets" that are applied to the spatial coordinates (latitude, longitude and depth) of the earthquake hypocenter, inter-event time and magnitude. The strict meaning of them is different from the conventional wavelets, although appear some similarities between the two entities.

Analysing the evolution in time of these physical wavelets the author extracts some

properties that are extrapolated in time in order to make a prediction of time of occurrence, magnitude and location of the 1995 Kobe earthquake from previous data. I admit that the paper is not completely clear to me in some passages, and most of the cited references are not accessible (many of them are in Japanese, some are in proceedings of conferences; see below) so I had to base my understanding on this sole article.

This approach reminds me that of nonlinear forecasting approach in a reconstructed phase space for a chaotic time series (e.g. Farmer and Sidorovich, 1987; Barraclough and De Santis, 1997). However what is missing in this paper is the preliminary analysis of the possible chaotic properties of the time series, which is fundamental before to arrive to any conclusion. In particular, in the present case study the prediction is advanced by 19 events with respect to the impending retrospectively predicted earthquake, but this number could be misleading in case, e.g., the time window of predictability (the reverse of the Kolmogorov entropy) could be smaller than the corresponding time (see e.g., De Santis et al. 2010). Also some other entropic analyses (e.g. De Santis et al., 2011) could be of some help, because they can provide an indication of the complexity of the corresponding time series.

Finally, a case study alone cannot establish the strength of a method, that can be locally dependent, where the fluctuations in the results could be due to some local/regional tectonics or by chance. I would suggest to show some other case studies to support the most general finding of the application of the author's method.

In the present version the article cannot be published. In summary, it requires certainly a major revision, with more clarification in some passages, estimation of some entropic properties of the time series in the reconstructed phase space, and some other case studies to see similarities or differences.

Some other minor points.

Pag.8 Lines 8-9. "The NCI(m, 2s) is proportional to seismic activity. If it is large,

the activity is quiet.." From the second sentence it seems that NCI(m,2s) is inversely proportional to seismic activity.

Pag.8 Lines 15-16 "The AMR's in the large region generally start a few days before a large event occurs somewhere in the region as well as before a large aftershock occurs (Takeda, 2015)." Generally AMR does not start a few days before a large event but starts months or even years before (Mignan et al., 2007).

In the references some articles are not easily accessible. For instance:

Takeda, F. and Okada, S.: Time Series Analysis with Physical Wavelets, 20 http://adsabs.harvard.edu/abs/2001APS.MARX23005T, 2001.

when I attempt to reach this document I have the following message: No valid abstract selected for retrieval or not yet indexed in ADS

In addition all below references are in Japanese:

TEC21 website: Crustal movement that caused the 2011 M9 Event, http://www.tec21.jp/g_eq_tohoku_crust_m.htm, 2017a.

TEC21 website: The 2011 M9 Event and Earthquake Prediction, http://www.tec21.jp/News_EQ_forecasting_j.htm, 2017b.

TEC21 website: Cycle of strain energy density accumulation, http://www.tec21.jp/critical_cycles.htm, 2017c.

TEC21 website: Predictions and Diagnostics–Industrial Systems, http://www.tec21.jp/Indust_sys_j.htm, 2017d.

TEC21 website: Precursors and Predictions, http://www.tec21.jp/pr_CQK_CQT_model_1.htm, 2017e.

References indicated in my review but not present in the article under scrutiny

Barraclough D. R. and A. De Santis, Some possible evidence for a chaotic geomagnetic

field from observational data, PEPI, 99, 207-220, 1997.

De Santis A., Cianchini G., Qamili E., Frepoli A.. The 2009 L'Aquila (Central Italy) seismic sequence as a chaotic process, Tectonophysics, 496 44–52, 2010.

De Santis A., Cianchini G., Beranzoli L., Favali P., Boschi E., The Gutenberg-Richter law and Entropy of earthquakes: two case studies in Central Italy, BSSA, v.101, 1386-1395, 2011.

Farmer J.D. and Sidorovich J.J., Predicting chaotic time series, Phys. Rev. Lett., 59. 845-848, 1987.

---

## Author Comment (AC1) · 10 Apr 2018

Replies to Referee #1

Specific comments:

(a) The present study for the Prediction of fault size, motion, magnitude and rupture time must be extended for all the cases with a magnitude over a specific threshold for the better evaluation of the suggested method.

Reply to (a):

The successful evaluation of our deterministic prediction model is made for the events whose M are larger than about 6 throughout Japan except for the motion analyses by using JMA unified hypocenter catalogs. For example, one evaluation has 15 cases presented at:

Takeda, F., The precursory fault width formation and critical stress state of impending large earthquakes: The observation and deterministic forecasting; AGU, Fall Meeting 2009, NH13A-1126, http://adsabs.harvard.edu/abs/2009AGUFMNH13A1126T

This reference should be added to the current article in relation to referrer #2's comments.

Reference (Takeda, 2015) has other events included. One of them is the 2011 great Tohoku EQ. The reference is the 130 page Japanese patent with 85 figures most of which are those figures like Fig. 2a-c, 3.a, 5, 6, 7a-b and 8a-b.

(b) The present study must be framed by the appropriate statistical analysis of the results including the false alarm rate.

Reply to (b):

One of our objectives is to show how to extract deterministic physical laws for precursory phenomena of impending large or great events by using a mathematical tool (physical wavelets). The extraction then allows us to build the physical models for CQK and CQT by which to predict impending large events. A few cases, which are related to EQ swarm and CQK stress loading, require some refinements on the deterministic prediction model as stated in the text. Since the model is based upon physical laws, the refinements are supported by physics as described in (Takeda, 2015). Our prediction model does not have a false alarm rate as the statistical prediction model should have.

(c) This analysis is closely related to the natural time analysis in which the order of the event (as an index) is also considered as one of the main characteristics of the examined time series. I am suggesting the following two references to be included: Natural-time analysis of critical phenomena: The case of seismicity PA Varotsos, NV Sarlis, ES Skordas, S Uyeda, M Kamogawa EPL (Europhysics Letters) 92 (2), 29002 Natural time analysis of critical phenomena P Varotsos, NV Sarlis, ES Skordas, S Uyeda, M Kamogawa Proceedings of the National Academy of Sciences 108 (28), 11361-11364

Reply to (c):

The natural time becomes a probabilistic quantity in its statistical model describing one aspect of seismicity. The references suggested by referee #1 have the seismicity in EQ catalogues studied within the framework of the statistical model studied in other branches of Statistical Physics.

The use of the chronological index as a time in the time series analyses has been established for many decades. The index does not become a probabilistic quantity so that one can study the statistical properties of the observations (time series data) as those (including DNA sequence in page 15) in 'Fractal Concepts in Surface Growth / A.-L. Barabási and H. E. Stanley'; Cambridge University Press 1995, and those in 'Is the Normal Heart Rate Chaotic?'; Chaos 19, 028501, 2009. Their studies are to find statistical quantities to characterize their physical systems.

In our EQ source parameter time series $d(c, m)$, where $c = LAT, LON, DEP, INT$ and $MAG$, the chronological index (time $m$ or $t$ ) is not a probabilistic quantity so that we can define time derivatives of $d(c, m)$ to find physical laws for precursory phenomena of impending large EQ.

In the seismic observation of $d(c, m)$, the virtual particle can change discontinuously in direction and speed just like a small particle (a colloidal particle) immersed in a large volume of liquid (Disperse systems / Makoto Takeo; Wiley-VCH, 1999, ISBN 3-527-29458-9, page 43 - 46). Thus it is not differentiable with respect to time.

Physical wavelets solve this issue of finding velocity and acceleration of the particle so that one can find physical laws for precursory phenomena of impending large EQs. The laws build a deterministic physical model of EQ prediction stated in section 4.2. The model can also be compared with other seismic (seismogenic) observations made by Jin and Aki as stated in the text. Our model, of course, must be refined by the observations to be made at various tectonically active regions.

Thus, the index time in time series data is not closely related to natural time in both many established statistical analyses and our deterministic analyses.

In the time series $d(c, m)$ of the daily displacement observed by GPS as stated in the text, the index $m$ is a day. The environmental noises of GPS prohibit time derivatives of $d(c, m)$; however, physical wavelets solve this issue. If there exists unique relationship between the real time and the index m, any index may be used as a time for its time derivatives of non-differentiable time series. Natural time cannot be used for the differentiations (see page 45 in Disperse systems / Makoto Takeo; Wiley-VCH, 1999).

(d) The analysis of the "Automatic detection of anomalies leading to the catastrophes of physical systems" needs improvement including for example among others a cross-correlation diagram respect to the time.

Reply to (d):

The examples of the cross correlation diagrams are given in references (Takeda, 1994, 1995, 1996;

Takeda et al., 2000; TEC21, 2017d). I can also add one of my Japanese patents, (Takeda, F.; Detecting Systems of Changes in Motion, Japanese Patent 2787143, J–PlatPat, JP, 1995-146161, A.1998), which was used for my consultant works of the industrial systems at large heavy industrial companies. The contracts prohibit any disclosure of detailed information. The principles and simulated experimental tests used in these projects are in the references stated above. I should have placed the references stated above in appropriate places.

Technical corrections:

(a) The quality of the figures needs improvement

Reply to (a);

I plan to improve the figure quality as much as I could.

(b) Grammatical errors:

Reply to (b):

I very much appreciate your corrections.

   Page 1, line 26: with Global -> by Global

   Page 1, line 30: on crustal surface -> on the crustal surface

   Page 2, line 4: As an example of many test -> As an example of many tests

My correction to this is:

As an example of many hindsight and real-time extractions

   Page 4, line 23: oscillometeric -> oscillometric C2

   Page 4, line 26: complete -> a complete

   Page 6, line 26: Similarly -> Similarly,

   Page 7, line 31: reginal -> regional

   Page 8, line 4: Similarly -> Similarly,

   Page 8, line 14: every large and great events -> every large and great event

   Page 10, line 2: Similarly -> Similarly,

   Page 10, line 14: by arrow -> by an arrow

   Page 11, line 16: and main event, -> and main event

   Page 11, line 17: The sequence appears as a cycle of strain energy accumulation and release to
   the M9 event, -> release: wrong grammar

My correction to this is:

The sequence appears as a cycle of strain energy accumulation and decrease to the M9 event,

   Page 11, line 24: characterized with magnitude -> characterized by magnitude

   Page 12, line 12: about a year and half -> about a year and a half

   Page 12, line 19: which appear -> which appears

   Page 12, line 28: prediction on the fault size and motion -> prediction of the fault size and motion

---

## Author Comment (AC2) · 10 Apr 2018

Replies to Referee #2

It is unfortunate that referee #2 was unfamiliar with the physical wavelet methodology used to find physical laws for the precursory phenomena of impending large EQ events. Therefore referee #2 has some difficulty understanding their relations to the seismogenic processes of impending large EQ events observed by seismologists as stated in section 5.

First, I would like to reply to your main objection by using the same reference stated in the reply to referee #1's section (a), which discusses deterministic chaos on the observed seismicity as in Fig. 3a. I would also like to place my replies following each of your some other minor points.

This work is not related to any of those statistical analyses suggested for proving the predictability. I may only say the following:

Reference (Takeda, F., The precursory fault width formation and critical stress state of impending large earthquakes: The observation and deterministic forecasting; AGU, Fall Meeting 2009, NH13A-1126, http://adsabs.harvard.edu/abs/2009AGUFMNH13A1126T), has the largest Lyapunov (Ly) exponents calculated by using the same time series as $d$ ($c$, $m$) in Fig. 3a, as listed in a table below. The reference shows that the largest exponents of $d$ ($c$, $m$) in the property space are all positive, statistically distinct from those surrogated by randomly shuffling only the event index. The t-test (for six surrogated data sets) with the confidence level of 99.9 % suggests that the seismicity of $d$ ($c$, $m$) is a deterministic chaos so that one may have a deterministic model for the prediction. However, there are three large EQs and one EQ swarm as in Fig. 3a (see also Fig. 2a). Since the Hurst exponents of each $d$ ($c$, $m$), which are also listed below, suggest that each has a long memory of large evets, the seismicity after main events is deterministic chaos. If one can find the same result before main events, one may be then able to make a physical model for the EQ predations. This evaluation of the deterministic chaos is difficult for any tectonically active region where large EQ evets often occur.

The statistical quantiles in the reference are as follows.

| c | Largest Ly exponent | t-test 99.9% level | Hust Exponent |
|---|---|---|---|
| LAT | 0.331 ± 0.028 | 0.410～0.451 | 0.84 |
| LON | 0.459 ± 0.032 | 0.497～0.551 | 0.77 |
| DEP | 0.335 ± 0.026 | 0.341～0.453 | 0.76 |
| INT | 0.241 ± 0.024 | 0.247～0.327 | 0.78 |
| MAG | 0.155 ± 0.019 | 0.156～0.199 | 0.58 |

One may note that the well-known Hurst exponents are 0.78 ±0.09 and 0.5 for many natural systems and Brownian motion, respectively.

Physical wavelets are fundamentally different as stated in the text (see also the replies to referee #1). I would like to make very simple statements here. The well-known wavelet analyses cannot obtain

velocity (momentum) and acceleration (force) from non-differentiable time series. Physical wavelets can find the same results obtained by the well-known wavelet analyses (Takeda, 1994). Similarly, time delay embedding cannot construct displacement-momentum or displacement-force phase spaces.

Some other minor points.

Pag.8 Lines 8-9. "The NCI(m, 2s) is proportional to seismic activity. If it is large, the activity is quiet.." From the second sentence it seems that NCI(m,2s) is inversely proportional to seismic activity.

Reply to this:

We discuss if the activity is quiet (seismic quiescence has large *INT*) or active (small *INT*). In this sense, large or small $NCI\,(m,\,2s)$ is quiet or active (or noisy), respectively.

Pag.8 Lines 15-16 "The AMR's in the large region generally start a few days before a large event occurs somewhere in the region as well as before a large aftershock occurs (Takeda, 2015)." Generally AMR does not start a few days before a large event but starts months or even years before (Mignan et al., 2007).

Reply to this:

The tool of (Mignan et al., 2007) to detect AMR is completely different from our tool to monitor the state of strain energy density expressed by NCI(m,2s) and NCD(m,2s). If the observational window of AMR is different, the start of AMR is different.

As an example of this, we have demonstrated that the observed periodicity of seismicity in section 5.1 will depend on the size of seismic region (in section 4.6), the observational tool and its width of time window.

As for this and my first reply stated above, a great mentor and seismologist Keiiti Aki, had commented in his email sent to me as follows:

Jan 27, 2005

Dear Takeda-san:

I finished reading through all the documents you sent me, and I now feel that our accidental meetings might have been planned by someone in the heaven.

Your successful accomplishment as an engineering consultant, your background as a physicist and your humanitarian wish to mitigate the earthquake disaster meet everything I expect for someone to practice the future earthquake prediction research. As I explained in my extended abstract of a talk at an international meeting on "Imaging Technology" held in Sendai last November, which I have asked Anshu Jin to mail to you, I believe that the earthquake prediction is not an academic problem, but an engineering application of Seismology. The problem involves three elements; physics, nature and society (the three most beautiful things human beings experienced in this world). I think we need someone like you to solve this problem.

I attended an international meeting in Spain in October, 2004, celebrating the centennial

anniversary of an old observatory. Don Turcotte was there and gave a talk concluding that the seismicity is a chaotic noise. I started my talk saying "I accept that completely as phenomena originating from the brittle part. Our data are dominated by the events from the brittle part. We need to find faint signals from the ductile part which can be modeled deterministically."

Here I recognize some difference between you and me. You are characterizing the phenomena as "deterministic chaos", opposing the view of Turcotte's group. It seems to me that there is no need for this conflict, if you consider that the physical system is not just the brittle part but includes the ductile part. I learned it from my experience with an active volcano as described in my Trieste lecture note, which again I asked Anshu to mail to you.

Thank you for an exciting time I had since reading your mails.

With best regards,

Kei Aki

Jan 27, 2005

Dear Takeda - san:

My excitement continues from reading your paper.

First, you do not seem to be bothered by the 60 - event periodicity, attributing it to some process at the brittle-ductile transition zone. Seismologists would react with the suspicion that some artifact in analysis causing it, and discredit your finer interpretation as your imagination. I am amazed in your confidence as a physicist that such fluctuation can be expected as a physical phenomenon. Personally I believe that this periodicity is real, indicating a clear departure of the process involved from the self-similarity, possibly due to the unique size of the fractures in the brittle part of the lithosphere (a few hundred meters to about a km) that I have proposed since the 1989 JGR paper with Anshu Jin. There are numerous observations supporting the existence of such a unique length as I described in my Trieste lecture note, but I still cannot prove it. For example, as you find in the fluctuation of coda Q and N(Mc) in California by Jin and Aki (as quoted in my 2004 EPS paper), we saw a periodicity of about 10 years. The fluctuations in these parameters in other areas are usually several years, much longer than what you showed in your figures. So there must be some artifact in the apparent periodicity that needs to be clarified before convincing seismologists about their physical reality.

Secondly, your distinction of CQT (T for Tottori) and CQK (K for Kobe) is extremely interesting because the high resolution map of coda Q obtained from the 1000 Hi-net stations and the map of N(Mc) from the JMA data both obtained recently by Anshu also identify the two areas not only as anomalous, but also in distinctly different ways. I have not digested fully these observations, but I feel that both you and Anshu are detecting the common phenomenon through different windows. Would you two exchange papers and start communicating each other? There is not much time left, because Anshu must quit her position at NIED at the end of March, as I mentioned in my earlier mail.

Have you read the extended abstract of my paper titled --A perspective on engineering application of

seismology-- presented at an international symposium organized by the Society of Exploration Geophysicists (SEG), Japan, which I asked Anshu to mail a copy? I have a feeling that my dream about the future of earthquake prediction described in that paper may be realized by you. Perhaps that was the intension of someone in the heaven who arranged several accidental meetings between you and me!

With best regards,

Kei

As for the periodicity of several to 10 years mentioned above, we have clarified in section 5.1 that our periodicity of about 2 years (about 600 days) becomes 8 years if we use his observational time window.

In the references some articles are not easily accessible.

For instance: Takeda, F. and Okada, S.: Time Series Analysis with Physical Wavelets, 20 http://adsabs.harvard.edu/abs/2001APS.MARX23005T, 2001. when I attempt to reach this document I have the following message: No valid abstract selected for retrieval or not yet indexed in ADS

Reply to this:

I appreciate your complete checkups of references. It is a typo. Two dots should be after APS.

http://adsabs.harvard.edu/abs/2001APS..MARX23005T

In addition all below references are in Japanese:

TEC21 website: Crustal movement that caused the 2011 M9 Event,

http://www.tec21.jp/g_eq_tohoku_crust_m.htm, 2017a.

TEC21 website: The 2011 M9 Event and Earthquake Prediction,

http://www.tec21.jp/News_EQ_forecasting_j.htm, 2017b.

TEC21 website: Cycle of strain energy density accumulation,

http://www.tec21.jp/critical_cycles.htm, 2017c.

TEC21 website: Predictions and Diagnostics–Industrial Systems,

http://www.tec21.jp/Indust_sys_j.htm, 2017d.

TEC21website: Precursors and Predictions,

http://www.tec21.jp/pr_CQK_CQT_model_1.htm, 2017e.

Reply to this:

They are all from my patents most of which are in Japanese. Their English translations are under way by the Japanese Patent office. Google may translate them for you. I am also writing a few English drafts to be submitted to some geophysical journals.

---

## Author Comment (AC3) · 21 Apr 2018

Final responses to referees #1 and #2

I experienced the referee process for Chaos of AIP so I know how much time referees have to spend in reviewing articles. I very much appreciate referee's time and comments.

It is, however, unfortunate that two referees were unfamiliar with the physical wavelet methodology used to find the deterministic physical laws for the precursory phenomena of impending large earthquake (EQ) events. They appear to confuse the deterministic methodology with statistical methods like natural time and chaos analyses. They also appear unfamiliar with the large earthquake phenomena observed in both seismicity and crustal displacements of GPS stations.

Clarifying some mix-ups on the methodologies applied to seismic observations may require a summarization as follows:

1) As stated in the introduction, both EQ events and the daily displacements of GPS stations quantify the stress state of the earth's lithosphere, which is deterministically controlled by tectonic forces.

2) The notion of a virtual particle for EQ events in the EQ source parameter space has been introduced since 2001 to find the deterministic physical laws for stress state that creates $d(c, m)$ where $c = LAT$, $LON$, $DEP$, $INT$ and $MAG$, and time $m$ is the chronological event index that is uniquely related to origin time (real time).

3) The chaotic motion of the particle (shown in Figure 3a) has periodic fluctuations of about 64 events common to all $c$ observed as in Figure 4. Their spectra are obtained from $d(c, m)$ whose time $m$ runs from 1986 to the 1995 Kobe EQ.

4) The $d(c, m)$ is non-differentiable with respect to time $m$ so that one needs a new mathematical tool of calculus to find the time derivatives.

5) The concise description of the tool with a mathematical proof is given in section 3. I have developed and used the tool for my private consulting works since 1985 as stated in sections 3.4 and 3.8.

6) The tool, named as physical wavelets, finds displacement $D(c, t)$, velocity $V(c, t)$ and acceleration $A(c, t)$ of the particle motion whose pathway is non-differentiable $d(c, m)$. The $d(c, m)$ has the periodic fluctuation components of about 64 events. As stated in sections 3.5 and 4.2, they are the Newtonian equations of particle motion in specific periodic fluctuations, which was masked by chaotic seismic noise.

7) Two types of anomalous accelerations, which are characterized by the phase relationship of $A(c, t)$ among $c = DEP$, $INT$ and $MAG$, are precursory to every large impending event with M $\geqq$ about 6 throughout Japan. One is named as CQK after the 1995 Kobe M7.2 and another is CQT after the 2000 Tottotri M7.2.

8) CQK and CQT are the deterministic physical laws for precursory phenomena of impending large EQs, which then build the physical models of deterministic EQ predictions.

9) The CQK and CQT are in perfect harmony with other seismogenic observations of coda waves as stated in section 5.

10) Keiiti Aki had personally commented on the CQK and CQT which are the foundation of deterministic EQ prediction model as quoted in the response to referee #2.

11) Similarly, as stated in section 3.7, we have analyzed the daily displacements of GPS stations of $d$ $(c, m)$ at time (day) $m$, where $c$ is the geological axis $E$ (west–east), $N$ (south–north) and $h$ (down–up). The $d$ $(c, m)$ has various trends and $d$ $(h, m)$ is completely masked by various environmental noises. So, $d(c, m)$ is non-differentiable. Physical wavelets find displacement $D(c, t)$, velocity $V(c, t)$ and acceleration $A(c, t)$ in selective frequency (being $1/m$), which are used for the quantitative analyses of the crustal motion in $D(c, t) - V(c, t)$ and $D(c, t) - A(c, t)$ phase planes. For example, the trajectory drawn on $D(h, t) - V(h, t)$ plane with $w = 200$ days and $2n = s = 300$ days, has the resolutions of 0.1 mm and 0.0001 mm/day 5 orders of magnitude greater than the daily background noise level. Thus they have identified the precursory crustal motion to the great Tohoku M9 event.

12) Physical wavelets can be used for the statistical analyses of time series data. Its simplest application is given in section 3.9 to find the strain energy density of a regional seismicity. The rapid release of energy is identified as the phenomenon known as AMR. It is shown it can be used as a prediction of an impending large event as shown in section 4.6.

13) As for the comments that more case studies are needed, they are from the statistical point view of a false alarm rate. Any deterministic analysis of seismic fault motion and structure should be made only for a single event as seismic moment is formulated by Keiiti Aki in 1966. The fault plane may not be single (simple) like that of the 1995 Kobe EQ but complex as observed in many cases. Inclusion of this kind of case study requires a book size document with many references like my patent (Takeda. F., 2015) and a life time to write. We have tested many cases successfully for our physical model as stated in section 4.3 by using the empirical relations of fault size and magnitude.

In closing my summary, I would like to stress that physical wavelets have deterministic and statistical methodologies. The deterministic methodology has a profound application to the calculus of non-differentiable deterministic and stochastic processes as applied to our finding of the deterministic physical laws for precursory phenomena for large and great impending EQs.